# Assessment of NDVI Dynamics of Maize (*Zea mays* L.) and Its Relation to Grain Yield in a Polyfactorial Experiment Based on Remote Sensing

**András Tamás, Elza Kovács, Éva Horváth, Csaba Juhász \*, László Radócz, Tamás Rátonyi and Péter Ragán**

Institute of Land Use, Engineering and Precision Farming Technology, Faculty of Agricultural and Food Sciences and Environmental Management, University of Debrecen, H-4032 Debrecen, Hungary
* Correspondence: juhasz@agr.unideb.hu

**Abstract:** Remote sensing is an efficient tool to detect vegetation heterogeneity and dynamics of crop development in real-time. In this study, the performance of three maize hybrids (Fornad FAO-420, Merida FAO-380, and Corasano FAO-490-510) was monitored as a function of nitrogen dose (0, 80 and 160 kg N ha⁻¹), soil tillage technologies (winter ploughing, strip-tillage, and ripping), and irrigation (rainfed and 3 × 25 mm) in a warm temperature dry region of East-Central Europe. Dynamics of the Normalized Difference Vegetation Index (NDVI) were followed in the vegetation period of 2021, a year of drought, by using sensors mounted on an unmanned aerial vehicle. N-fertilization resulted in significantly higher NDVI throughout the entire vegetation period ($p < 0.001$) in each experimental combination. A significant positive effect of irrigation was observed on the NDVI during the drought period (77–141 days after sowing). For both the tillage technologies and hybrids, NDVI was found to be significantly different between treatments, but showing different dynamics. Grain yield was in strong positive correlation with the NDVI between the late vegetative and the early generative stages (r = 0.80–0.84). The findings suggest that the NDVI dynamics is an adequate indicator for evaluating the impact of different treatments on plant development and yield prediction.

**Keywords:** unmanned aerial vehicle; growth dynamics; yield–NDVI correlation; polyfactorial experiment; remote sensing

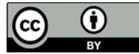

## 1. Introduction

Field crop production is of paramount importance; it is an essential basis for food production and a source of our existence. Maize is one of the most significant crops in the world; it is necessary to provide both food and energy security [1,2]. Hungary covers a total area of 9.3 million hectares, of which 4.1 million hectares are arable land. Soil moisture is the most direct and important source of water demand for crops [3]. Therefore, climate change is the largest challenge of our time, resulting in less and less precipitation in Hungary every year [4]. Drought is currently a severe environmental limiting factor of crop survival and productivity due to the continuous decline in global water resources [5]. The increasing drought causes severe damage to agriculture, especially crop production [6,7]. According to Noromiarilanto et al. [8], the situation will undoubtedly deteriorate as climate change estimates show an increase in extreme weather events.

The long-term impacts of yield loss are crucial for socioeconomic stability and food security [9]. Research by Horváth et al. [10] showed that the amount of N fertilizer and application time recommended to achieve the highest yields varies among hybrids with different FAO numbers, which is also influenced by environmental factors.

Maize is cultivated in many countries around the world [11] and it is an important source of food, animal feed, and fuel [12]. In our country, the most dominant arable crop

is maize (Zea mays L.) [13], the cultivation of which is highly dependent on climate. In addition to the season, fertilization [14] and the applied hybrid [15] have a significant influence on maize yield. The research of Rácz et al. [16] highlighted, in addition to the environmental conditions during the season, the need for proper N ha$^{-1}$ treatment as it significantly affects maize biomass yield and increases the protein content of the crop.

Global agricultural production is extremely sensitive to the negative impacts of climate change (extreme weather, drought, heatwaves), and Hungary is no exception [17]. Crops react differently; they have different responses to abiotic and biotic environmental impacts and, obviously, to other treatments. ML models could offer an appropriate scope for analyzing the vast amounts of data. Agriculture may soon adopt ML technology on a regular basis for tasks such as stress detection, yield prediction and estimation, and real-time field operations [18,19].

The interaction of light and foliage is determined by the physical and chemical properties of vegetation that affect the absorption, transmission and reflectance of light [20], and these create unique spectral properties for different species and phenotypes [21].

To record the different responses, we used UAV remote sensing to record Normalized Difference Vegetation Index (NDVI) values at different measurement times. Plant biomass, developmental stage, and environmental and other stress responses influence leaf spectral values, thus providing a good indication of the current state of the plant. Verhulst et al. [22] found a strong correlation between NDVI values and the accumulation of biomass both in maize and wheat. Remote-sensing-based plant studies, including UAV-based plant studies, are an excellent tool for assessing maize health and the response of different hybrids to treatments [23]. Remote sensing can be used to cover large areas quickly and efficiently.

The NDVI provides producers with the opportunity to assess crop biomass and yield. These indirect reflectance measurements have been used to estimate plant biomass and yield [24].

UAVs can be used to observe vegetation heterogeneity and its dynamic temporal variation [25–27]. These unmanned aerial vehicles are increasingly used for crop surveys [28,29].

Research by Burke and Lobell [30] suggested that NDVI values in the range 0.14-0.88 can be used for yield mapping. The strong (r = 0.89) NDVI - yield correlation was confirmed by other research [31,32], and can be used to monitor production.

The climate of Europe and our country has undergone significant changes in the last 30 years, but especially in the last 4–5 years. The average monthly temperature is rising, and the spatial and temporal distribution of precipitation is becoming more and more unfavorable at the European level. The long-term experiment is a reference for the production practice. The vegetation dynamics of different maize hybrids are different in every geographic area; therefore, it is necessary to narrow down the spatial extent for a more precise examination. UAV-based remote sensing is more accurate and reliable than satellite-based surveys because there are fewer peripheral disturbing factors. In this way, the tests can be better adapted to the given phenological phases.

NDVI measurements based on remote sensing at different times during the vegetation period can be used to predict the expected yield.

## 2. Materials and Methods

### 2.1. Experimental Location and Setup

The study was carried out in Hungary, at the Látókép Experimental Station of the University of Debrecen (N 47°33' E 21°27'), on calcareous chernozem soil. The complex soil tillage experiment (rotation x tillage x fertilization x irrigation x plant density x genotype) was set up in 1989, which is unique in the country and in Europe (Figure 1) [33,34]. The tillage block of the experiment is 8064 m², divided into one irrigated and one non-irrigated block. The main plot size is 2688 m², while the size of the fertilizer treatment plots

is 336 m² in total. The trials were conducted with three maize hybrids, each with different genotype (Merida FAO-380; Corasano FAO-490-510; Fornad FAO-420) in the crop year of 2021. The sowing date was 22 April, and the first day of emergence was 4 May. The number of plants was 80,000 plants ha⁻¹. A total of 30 kg N ha⁻¹ and 100% of P and K were applied as a basal fertilizer in autumn. The additional N active ingredient was applied in spring as liquid Nitrosol (27% N + 2% S) by top-dressing in June. The irrigated part of the area received 25 mm of irrigation water in June and 25 - 25 mm in early and late July. In the experiment, the NDVI dynamics of maize were studied under three different tillage regimes: winter ploughing, strip-tillage, and ripping. Harvesting took place on 28 September 2021 with a yield-measuring plot harvester.

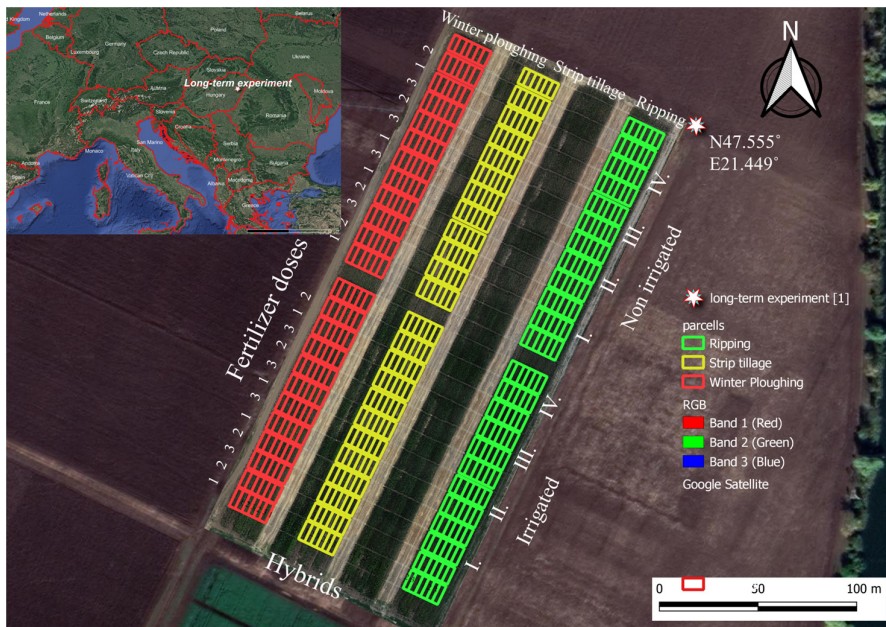

**Figure 1.** Field map with basemap of RGB image of 1st July 2021 and location in Hungary. Note: control, 80kg N ha⁻¹ + 60kg P₂O₅ ha⁻¹ + 90kg K₂O ha⁻¹, and 160kg N ha⁻¹ + 60kg P₂O₅ ha⁻¹ + 90kg K₂O ha⁻¹ fertilization treatments were randomized; I., II., III., IV: replications; three different tillage methods, inside the treatment blocks of the three maize varieties (1) "Merida FAO-380", (2) "Corasano FAO-490-510", and (3) "Fornad FAO-420".

### 2.2. Soil Characteristics of the Experiment

The soil type is chernozem, solonetzic in the deeper layers, which is typical of the Great Hungarian Plain, situated in the Carpathian Basin, East-Central Europe. Table 1 provides an overview of the soil characteristics.

**Table 1.** Main soil properties of the experimental area (Debrecen-Látókép, 2021).

|  | Layer 0–20 cm | Layer 20–40 cm | Layer 40–60 cm |
|---|---|---|---|
| pH (KCl 1:2,5) | 7.44 | 7.50 | 7.75 |
| $K_A$ | 45.5 | 46 | 46 |
| CaCO₃ (%) | 12.12 | 12.32 | 17.37 |
| Humus (%) | 2.86 | 3.09 | 2.11 |
| NO₃ + NO₂ (mg kg⁻¹) | 5.07 | 3.53 | 2.77 |
| P O₂₅ (AL) (mg kg⁻¹) | 515.98 | 533.43 | 173.05 |
| K₂ O (AL) (mg kg⁻¹) | 351.73 | 300.97 | 174.24 |

Note: $K_A$: Arany's plasticity index; AL: ammonium lactate-soluble.

### 2.3. Weather Characteristics of the Crop Season

The weather station of the experimental site measures air temperature by means of a P100 plantinum resistance thermometer, global radiation is measured by a Kipp & Zonen SP Lite2 pyranometer, and precipitation is measured by a Hungarian brand PG200 rain gauge, which measures the gravimetric principle. Among the official Hungarian National Meteorological Service (OMSZ) measurements, the temperature and precipitation data available for the city of Debrecen are 30-year averages, and the solar radiation data provide a 20-year average [35].

Agrometeorological conditions for the 2021 year were generally unfavorable. The last third of June was more than 3°C warmer than the 30-year average, and this month is the generative phase of maize. The very hot weather was accompanied by a severe lack of precipitation. In June, 6.4 mm of rain fell in total, well below the 30-year average. The drought continued in July and August (Figure 2).

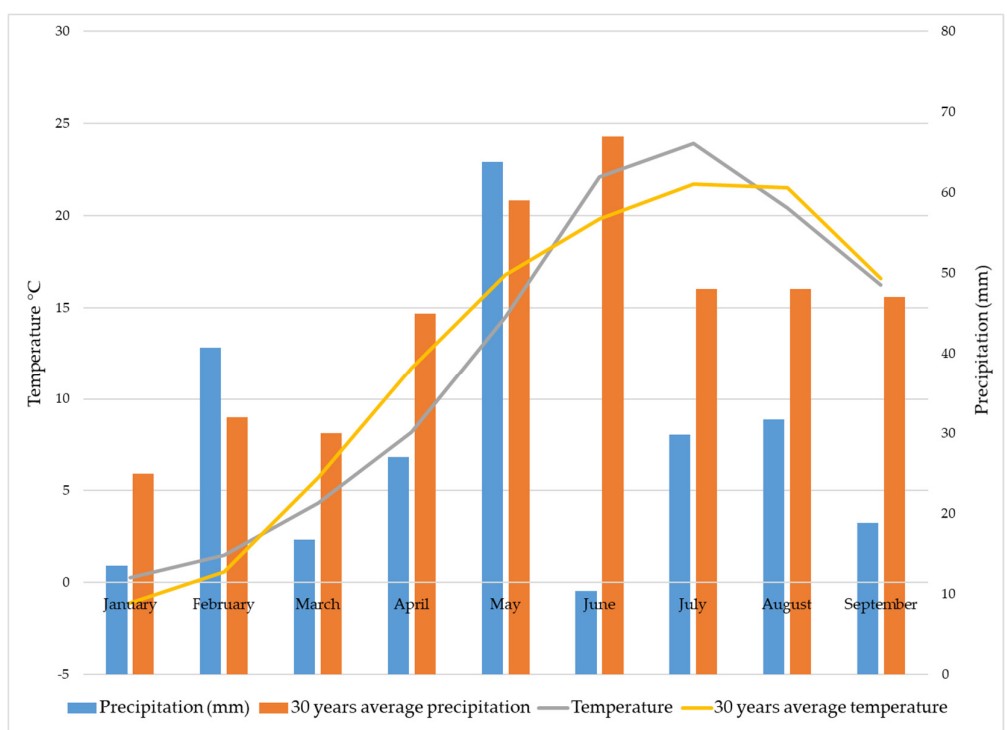

**Figure 2.** Monthly average air temperatures and monthly precipitation totals compared to 30–year average. The 30–year average: monthly average values for the last 30 years at the experiment site (Debrecen-Látókép, 2021).

The intensity of solar radiation during the emergence was around the 20-year average, but it exceeded the 20-year average by 23.5% in the intensive growth phase in June, and by 19.4% in the generative phase in July (Figure 3).

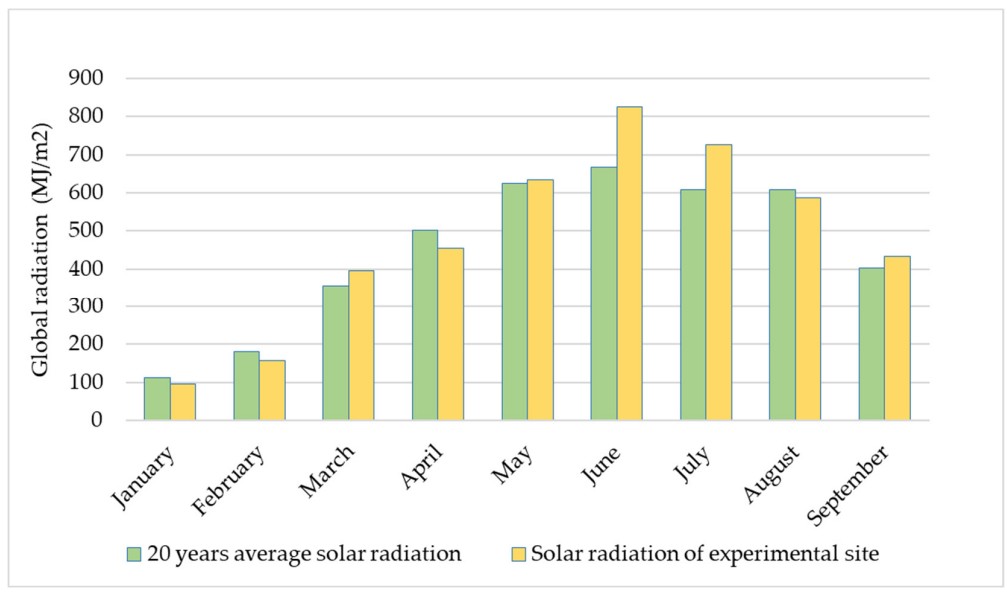

**Figure 3.** The 20–year average and 2021 monthly average of solar radiation at the experimental site (Debrecen-Látókép, 2021).

### 2.4. Methodology of Measurements and Calculations

NDVI$_{UAV}$ measurements were taken six times (days after sowing (DAS) 42; 56; 77; 90; 105; 141) using UAV. The DAS values correspond to the following phenological phases: V6; V8; V12; VT; R1; R3. Remotely sensed NDVI photos were captured using a DJI Phantom 4 Pro V2 drone with a Sentera Double 4K NDVI sensor. In QGIS 3.16.12, the orthorectified images were examined [36]. The methodology of the experiment was the same as conducted in a previously published article [37].

The WebODM software was used for stitching to complete NDVI images. DEM resolution was 2.0, DSM: true, DTM: true and the orthophoto spatial resolution was 2.0 cm.

As always, if the certified software of Sentera Inc. is not used, channels have to be further separated. The impact of out-of-band channels on each band can be removed using a set of equations. This calculation takes into account a roughly homogenous incoming light source (which is the case with daylight). The Sentera camera is a multispectral camera with a modified color filter, and the bands from the stored channels can be calculated using the formulae below [38]:

$$Red = -0.966 \times DNblue + 1.000 \times DNred \tag{1}$$

$$NIR = 4.350 \times DNblue - 0.286 \times DNred \tag{2}$$

$$NDVI = \frac{(NIR - Red)}{(NIR + Red)} \tag{3}$$

Trimble RTK was used to identify the edges of the plot. At different times, the generated polygon shape file was input into the stitched orthophoto. The raster was then updated using the formula by the QGIS program, and zone statistics were utilized to ascertain the mean values on each plot.

### 2.5. Data Analysis

Statistical evaluation was carried out using the R 4.2.2 statistical software environment [39]. The graphical interface was implemented using RStudio [40], gplots [41], car [42] and agricolae [43] packages. Graphs were created using Microsoft Excel. The first-

order error was set at 5%, i.e. alpha = 0.05. To investigate the effects of treatments, a repeated measures ANOVA model was constructed [44], and to compare the mean of yields, the least significant difference (LSD) approach was employed.

## 3. Results

The NDVI$_{UAV}$ values were significantly ($p < 0.001$) different between the different measurement times for hybrid and treatment averages. The highest mean value was measured on day 90 after sowing. The mean value measured on day 42 was 67.8% lower. On day 46 it was 59.9% and on day 56 45.8% lower. Day 77 and day 105 were 15.7–17.4% lower. On day 141 there was a significant 49% drop compared to day 90 (Figure 4).

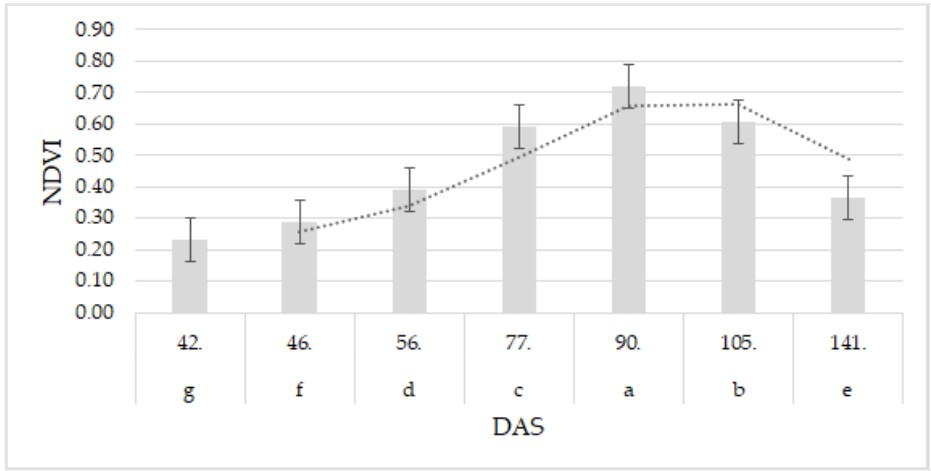

**Figure 4.** The NDVI$_{UAV}$ values of maize as the effect of days after sowing (DAS) (Debrecen, 2021); means of the varieties ± standard error. The differences among the measurement days were significant at the $p = 0.001$ level. Different letters mean significant difference at the $p < 0.05$ level among the measurement days. LSD = 0.003.

The correlation was inspected between the NDVI$_{UAV}$ values and yield t ha$^{-1}$ with Pearson correlation analysis which can be seen in Table 2.

**Table 2.** Pearson correlation coefficient (r) values among NDVI$_{UAV}$ and yield t ha$^{-1}$ (2021, Debrecen).

|  |  |  | 95% Cl | | |
| --- | --- | --- | --- | --- | --- |
| **DAS** | **Variables** | **Pearson's r** | **Lower** | **Upper** | **N** |
| 42 | NDVI$_{UAV}$ - Yield t ha$^{-1}$ | 0.420 *** | 0.339 | 0.494 | 432 |
| 46 | NDVI$_{UAV}$ - Yield t ha$^{-1}$ | 0.661 *** | 0.605 | 0.711 | 432 |
| 56 | NDVI$_{UAV}$ - Yield t ha$^{-1}$ | 0.676 *** | 0.622 | 0.724 | 432 |
| 77 | NDVI$_{UAV}$ - Yield t ha$^{-1}$ | 0.803 *** | 0.766 | 0.834 | 432 |
| 90 | NDVI$_{UAV}$ - Yield t ha$^{-1}$ | 0.821 *** | 0.787 | 0.849 | 432 |
| 105 | NDVI$_{UAV}$ - Yield t ha$^{-1}$ | 0.844 *** | 0.815 | 0.869 | 432 |
| 141 | NDVI$_{UAV}$ - Yield t ha$^{-1}$ | 0.577 *** | 0.510 | 0.637 | 432 |

***. Correlation is significant at the 0.001 level; confidence level (Cl) 95%.

It was expected to find a close correlation between the NDVI values and maize yield (Figure 5), and the coefficients pointed to very close positive connection between them at DAS 77, 90, 105 (r = 0.803–0.844), except the other measurement times, where the correlation was weaker (r = 0.420–0.676). The correlation was significant ($p = 0.001$).

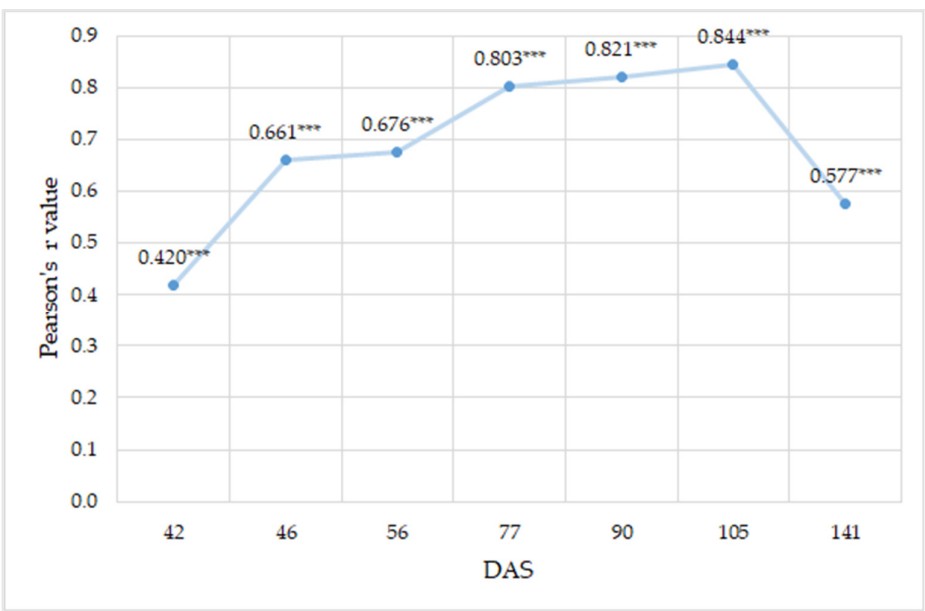

**Figure 5.** Pearson correlation coefficient (r) dynamic among NDVI$_{UAV}$ and yield t ha$^{-1}$ (2021, Debrecen). Note: ***. Correlation is significant at the 0.001 level.

When examining the effect of irrigation on NDVI$_{UAV}$ values measured at different times, there was a significant ($p < 0.001$) difference between the given DAS except for 42 DAS. Vegetative development up to 56 DAS was minimally lower in the irrigated sections by 1.1–7.9%. From 77 DAS, the irrigation effect was positive, 3.4–19.6%. The largest positive irrigation effect was at 77 DAS: +19.6% (Figure 6).

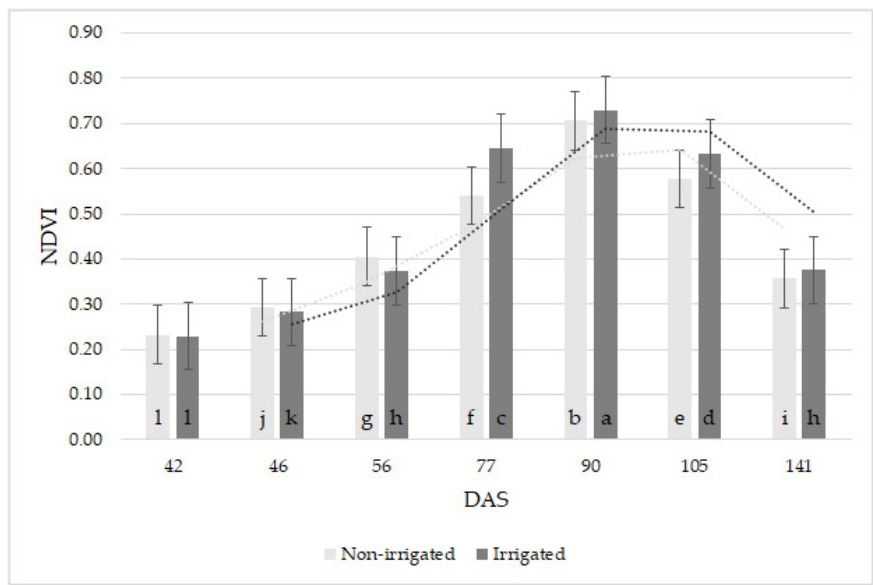

**Figure 6.** The NDVI$_{UAV}$ values of maize as the effect of irrigation at different DAS (Debrecen, 2021); means of the varieties ± standard error. At the $p = 0.001$ level, the differences between the DAS treatments were considered significant. The different letters indicate treatments that differ significantly ($p < 0.05$). LSD = 0.004.

Pearson's correlation coefficient trends were examined in both the non-irrigated and irrigated variants (Table 3).

**Table 3.** Pearson correlation coefficient (r) values among NDVI$_{UAV}$ and yield t ha$^{-1}$ in non-irrigated and irrigated conditions (2021, Debrecen).

| DAS | Group | Variables | Pearson's r | 95% Cl Lower | 95% Cl Upper | N |
|---|---|---|---|---|---|---|
| 42 | Non-irrigated | NDVI$_{UAV}$ - Yield t ha$^{-1}$ | 0.500 *** | 0.393 | 0.594 | 216 |
|  | Irrigated | NDVI$_{UAV}$ - Yield t ha$^{-1}$ | 0.437 *** | 0.322 | 0.539 | 216 |
| 46 | Non-irrigated | NDVI$_{UAV}$ - Yield t ha$^{-1}$ | 0.635 *** | 0.548 | 0.708 | 216 |
|  | Irrigated | NDVI$_{UAV}$ - Yield t ha$^{-1}$ | 0.760 *** | 0.697 | 0.811 | 216 |
| 56 | Non-irrigated | NDVI$_{UAV}$ - Yield t ha$^{-1}$ | 0.677 *** | 0.598 | 0.743 | 216 |
|  | Irrigated | NDVI$_{UAV}$ - Yield t ha$^{-1}$ | 0.813 *** | 0.762 | 0.854 | 216 |
| 77 | Non-irrigated | NDVI$_{UAV}$ - Yield t ha$^{-1}$ | 0.772 *** | 0.712 | 0.821 | 216 |
|  | Irrigated | NDVI$_{UAV}$ - Yield t ha$^{-1}$ | 0.893 *** | 0.862 | 0.917 | 216 |
| 90 | Non-irrigated | NDVI$_{UAV}$ - Yield t ha$^{-1}$ | 0.830 *** | 0.784 | 0.868 | 216 |
|  | Irrigated | NDVI$_{UAV}$ - Yield t ha$^{-1}$ | 0.858 *** | 0.818 | 0.889 | 216 |
| 105 | Non-irrigated | NDVI$_{UAV}$ - Yield t ha$^{-1}$ | 0.882 *** | 0.849 | 0.909 | 216 |
|  | Irrigated | NDVI$_{UAV}$ - Yield t ha$^{-1}$ | 0.860 *** | 0.820 | 0.891 | 216 |
| 141 | Non-irrigated | NDVI$_{UAV}$ - Yield t ha$^{-1}$ | 0.582 *** | 0.486 | 0.664 | 216 |
|  | Irrigated | NDVI$_{UAV}$ - Yield t ha$^{-1}$ | 0.547 *** | 0.446 | 0.634 | 216 |

***. Correlation is significant at the 0.001 level; confidence level (Cl) 95%.

Medium correlation was observed at early measurement dates (r = 0.437–0.635) and at the time of crop maturity (r = 0.547–0.582). From the 56th day of measurements, the start of actual irrigation, to the end of the vegetative stage (DAS 90), the correlation coefficient between NDVI$_{UAV}$ and yield t ha$^{-1}$ was higher due to the effect of irrigation (Figure 7).

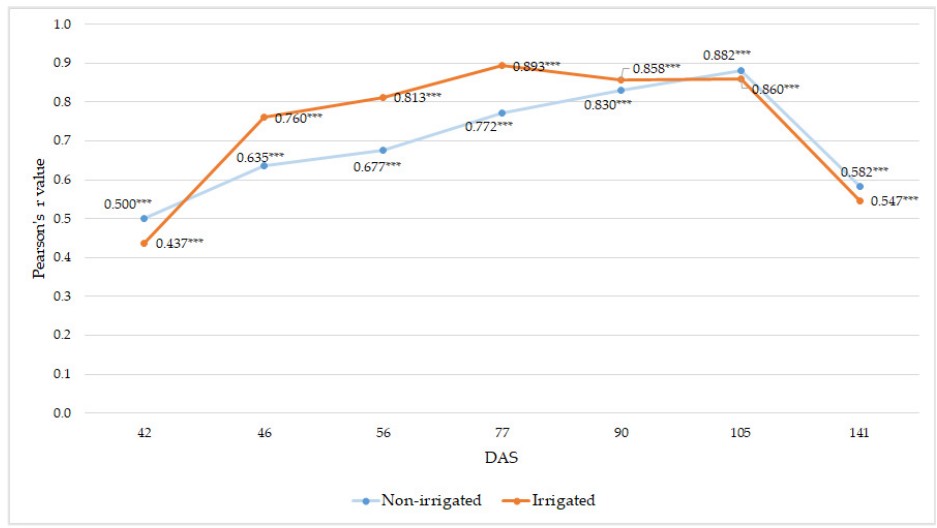

**Figure 7.** Pearson correlation coefficient (r) dynamics among NDVI$_{UAV}$ and yield t ha$^{-1}$ in non-irrigated and irrigated conditions (2021, Debrecen). Note: ***. Correlation is significant at the 0.001 level.

A significant difference is shown in Figure 8, when the effects of tillage types are examined. The tillage types differed at the $p < 0.001$ level at the time of measurement. Until 77 DAS, the maize crops after winter ploughing had the highest NDVI$_{UAV}$ values. The variance for strip tillage maize plots ranged from 4.9 to 17.9%, while the variance for ripping was lower with a negative variance of 1.8 to 4.5%. At 90 DAS, the highest NDVI$_{UAV}$ values were measured for ripping, with average values 0.3–4.3% lower in the winter ploughing and 1.4–6.1% lower in strip-tillage.

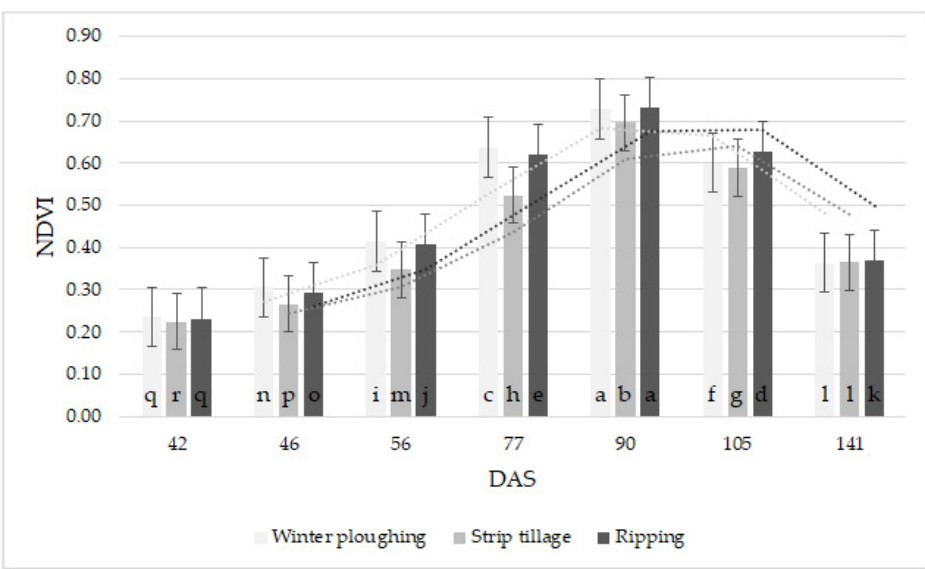

**Figure 8.** The NDVI$_{UAV}$ values of maize as the effect of tillage at different DAS (Debrecen, 2021); means of the varieties ± standard error. At the $p = 0.001$ level, the differences between the DAS treatments were considered significant. The different letters indicate treatments that differ significantly ($p < 0.05$). LSD = 0.005.

Correlation with yield varied across tillage treatments (Table 4).

**Table 4.** Pearson correlation coefficient (r) values among NDVI$_{UAV}$ and yield t ha$^{-1}$ as a result of tillage systems (2021, Debrecen).

| DAS | Group | Variables | Pearson's r | 95% Cl Lower | 95% Cl Upper | N |
|---|---|---|---|---|---|---|
| 42 | Winter ploughing | NDVI$_{UAV}$ - Yield t ha$^{-1}$ | 0.436 *** | 0.293 | 0.559 | 144 |
| | Strip-till | NDVI$_{UAV}$ - Yield t ha$^{-1}$ | 0.129 *** | −0.035 | 0.286 | 144 |
| | Ripping | NDVI$_{UAV}$ - Yield t ha$^{-1}$ | 0.368 *** | 0.218 | 0.502 | 144 |
| 46 | Winter ploughing | NDVI$_{UAV}$ - Yield t ha$^{-1}$ | 0.657 *** | 0.553 | 0.741 | 144 |
| | Strip-till | NDVI$_{UAV}$ - Yield t ha$^{-1}$ | 0.498 *** | 0.365 | 0.612 | 144 |
| | Ripping | NDVI$_{UAV}$ - Yield t ha$^{-1}$ | 0.650 *** | 0.545 | 0.736 | 144 |
| 56 | Winter ploughing | NDVI$_{UAV}$ - Yield t ha$^{-1}$ | 0.590 *** | 0.472 | 0.687 | 144 |
| | Strip-till | NDVI$_{UAV}$ - Yield t ha$^{-1}$ | 0.587 *** | 0.469 | 0.685 | 144 |
| | Ripping | NDVI$_{UAV}$ - Yield t ha$^{-1}$ | 0.719 *** | 0.630 | 0.790 | 144 |
| 77 | Winter ploughing | NDVI$_{UAV}$ - Yield t ha$^{-1}$ | 0.813 *** | 0.749 | 0.862 | 144 |
| | Strip-till | NDVI$_{UAV}$ - Yield t ha$^{-1}$ | 0.712 *** | 0.621 | 0.785 | 144 |
| | Ripping | NDVI$_{UAV}$ - Yield t ha$^{-1}$ | 0.815 *** | 0.751 | 0.863 | 144 |
| 90 | Winter ploughing | NDVI$_{UAV}$ - Yield t ha$^{-1}$ | 0.846 *** | 0.792 | 0.887 | 144 |
| | Strip-till | NDVI$_{UAV}$ - Yield t ha$^{-1}$ | 0.726 *** | 0.638 | 0.795 | 144 |
| | Ripping | NDVI$_{UAV}$ - Yield t ha$^{-1}$ | 0.869 *** | 0.823 | 0.904 | 144 |
| 105 | Winter ploughing | NDVI$_{UAV}$ - Yield t ha$^{-1}$ | 0.920 *** | 0.891 | 0.942 | 144 |
| | Strip-till | NDVI$_{UAV}$ - Yield t ha$^{-1}$ | 0.825 *** | 0.764 | 0.871 | 144 |
| | Ripping | NDVI$_{UAV}$ - Yield t ha$^{-1}$ | 0.858 *** | 0.808 | 0.896 | 144 |
| 141 | Winter ploughing | NDVI$_{UAV}$ - Yield t ha$^{-1}$ | 0.685 *** | 0.587 | 0.763 | 144 |
| | Strip-till | NDVI$_{UAV}$ - Yield t ha$^{-1}$ | 0.562 *** | 0.438 | 0.664 | 144 |
| | Ripping | NDVI$_{UAV}$ - Yield t ha$^{-1}$ | 0.624 *** | 0.513 | 0.715 | 144 |

***. Correlation is significant at the 0.001 level; confidence level (Cl) 95%.

At the early (DAS 42) measurement date, strip-tillage showed a weak correlation (r = 0.129) and a lower correlation between NDVI$_{UAV}$ and yield t ha$^{-1}$ at all measurement dates.

The strongest correlation was observed at 90 and 105 days after sowing, DAS 90 for ripping (r = 0.869), and DAS 105 for winter ploughing (r = 0.920) (Figure 9).

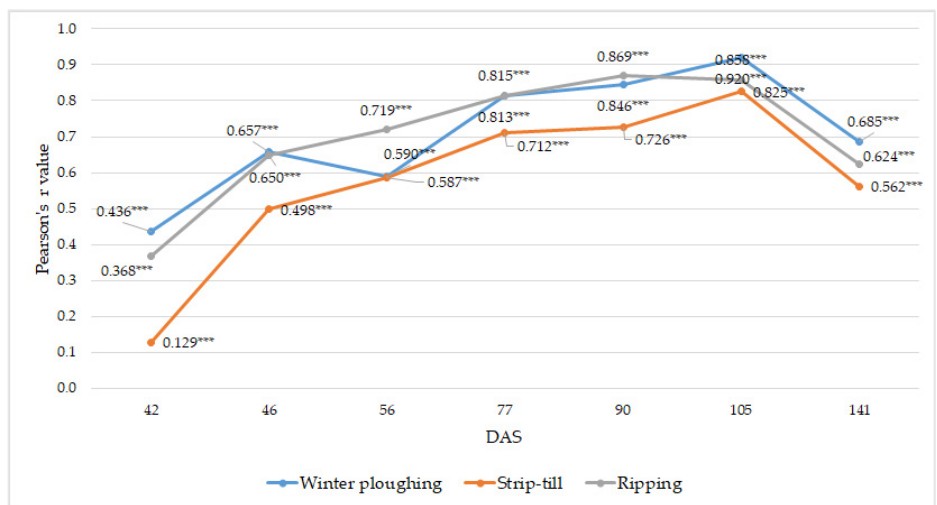

**Figure 9.** Pearson's correlation coefficient (r) dynamics among NDVI$_{UAV}$ and yield t ha$^{-1}$ as a result of tillage systems (2021, Debrecen). Note: ***. Correlation is significant at the 0.001 level.

Examining the NDVI$_{UAV}$ dynamics of the hybrids (Figure 10), the difference can be statistically confirmed ($p < 0.001$). The highest mean values were measured for the Fornad FAO-420 hybrid. Overall, there was no significant difference, with Merida FAO-380 having 0.2–11.2% lower NDVI at different time points, and Corasano FAO-490-510 having 0.3–10.5% lower NDVI at the last measurement time (141 DAS).

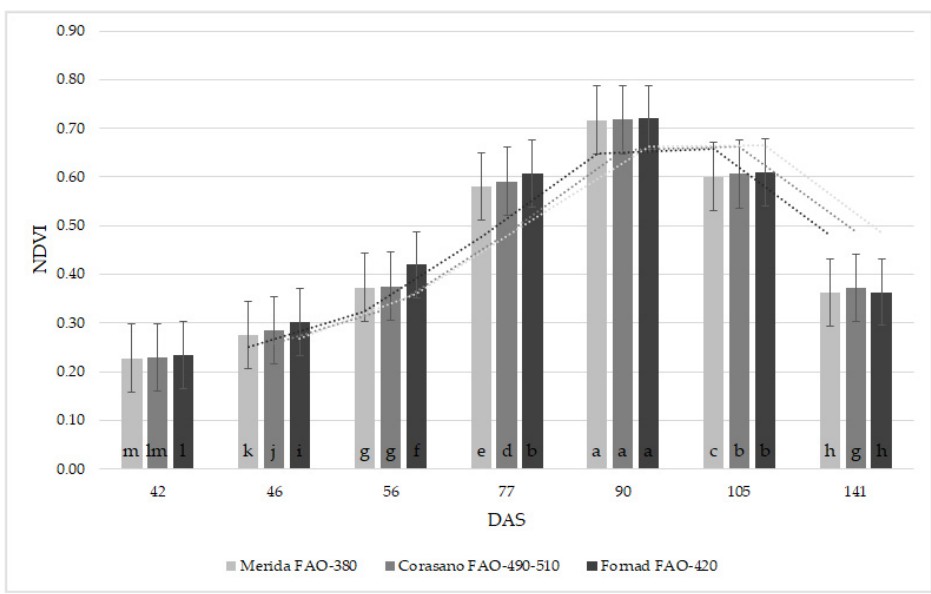

**Figure 10.** The NDVI$_{UAV}$ values of maize hybrids at different DAS (Debrecen, 2021); means of the varieties ± standard error. Differences between the examined hybrids at DAS were significant at the $p = 0.001$ level. Different letters indicate significant difference at the $p < 0.05$ level among the hybrids and DAS. LSD = 0.005.

The correlation between the NDVI$_{UAV}$ values of the hybrids and the yield t ha$^{-1}$ varied due to the genotypes (different FAO), as is shown in Table 5.

**Table 5.** Pearson correlation coefficient (r) values among NDVI$_{UAV}$ and yield t ha$^{-1}$ as a result of hybrids (2021, Debrecen).

| DAS | Group | Variables | Pearson's r | 95% Cl Lower | Upper | N |
|-----|-------|-----------|-------------|-------|-------|---|
| 42 | Merida-380 | NDVI$_{UAV}$ - Yield t ha$^{-1}$ | 0.436 *** | 0.293 | 0.559 | 144 |
| | Corasano-490-510 | NDVI$_{UAV}$ - Yield t ha$^{-1}$ | 0.129 *** | -0.035 | 0.286 | 144 |
| | Fornad-420 | NDVI$_{UAV}$ - Yield t ha$^{-1}$ | 0.368 *** | 0.218 | 0.502 | 144 |
| 46 | Merida-380 | NDVI$_{UAV}$ - Yield t ha$^{-1}$ | 0.657 *** | 0.553 | 0.741 | 144 |
| | Corasano-490-510 | NDVI$_{UAV}$ - Yield t ha$^{-1}$ | 0.498 *** | 0.365 | 0.612 | 144 |
| | Fornad-420 | NDVI$_{UAV}$ - Yield t ha$^{-1}$ | 0.650 *** | 0.545 | 0.736 | 144 |
| 56 | Merida-380 | NDVI$_{UAV}$ - Yield t ha$^{-1}$ | 0.590 *** | 0.472 | 0.687 | 144 |
| | Corasano-490-510 | NDVI$_{UAV}$ - Yield t ha$^{-1}$ | 0.587 *** | 0.469 | 0.685 | 144 |
| | Fornad-420 | NDVI$_{UAV}$ - Yield t ha$^{-1}$ | 0.719 *** | 0.630 | 0.790 | 144 |
| 77 | Merida-380 | NDVI$_{UAV}$ - Yield t ha$^{-1}$ | 0.813 *** | 0.749 | 0.862 | 144 |
| | Corasano-490-510 | NDVI$_{UAV}$ - Yield t ha$^{-1}$ | 0.712 *** | 0.621 | 0.785 | 144 |
| | Fornad-420 | NDVI$_{UAV}$ - Yield t ha$^{-1}$ | 0.815 *** | 0.751 | 0.863 | 144 |
| 90 | Merida-380 | NDVI$_{UAV}$ - Yield t ha$^{-1}$ | 0.846 *** | 0.792 | 0.887 | 144 |
| | Corasano-490-510 | NDVI$_{UAV}$ - Yield t ha$^{-1}$ | 0.726 *** | 0.638 | 0.795 | 144 |
| | Fornad-420 | NDVI$_{UAV}$ - Yield t ha$^{-1}$ | 0.869 *** | 0.823 | 0.904 | 144 |
| 105 | Merida-380 | NDVI$_{UAV}$ - Yield t ha$^{-1}$ | 0.920 *** | 0.891 | 0.942 | 144 |
| | Corasano-490-510 | NDVI$_{UAV}$ - Yield t ha$^{-1}$ | 0.825 *** | 0.764 | 0.871 | 144 |
| | Fornad-420 | NDVI$_{UAV}$ - Yield t ha$^{-1}$ | 0.858 *** | 0.808 | 0.896 | 144 |
| 141 | Merida-380 | NDVI$_{UAV}$ - Yield t ha$^{-1}$ | 0.685 *** | 0.587 | 0.763 | 144 |
| | Corasano-490-510 | NDVI$_{UAV}$ - Yield t ha$^{-1}$ | 0.562 *** | 0.438 | 0.664 | 144 |
| | Fornad-420 | NDVI$_{UAV}$ - Yield t ha$^{-1}$ | 0.624 *** | 0.513 | 0.715 | 144 |

***. Correlation is significant at the 0.001 level; confidence level (Cl) 95%.

In the initial development, the Corasano FAO-490-510 hybrid shows a weak correlation with yield (r = 0.129), which can be explained by its mid–late maturity (Figure 11). The hybrids show the strongest correlation at different times during the growing season: Merida FAO-380 at 105 days after sowing (r = 0.920), Fornad FAO-420 at 90 days after sowing (r = 0.869), and mid–late maturing Corasano FAO-490-510 at 105 days after sowing (r = 0.825).

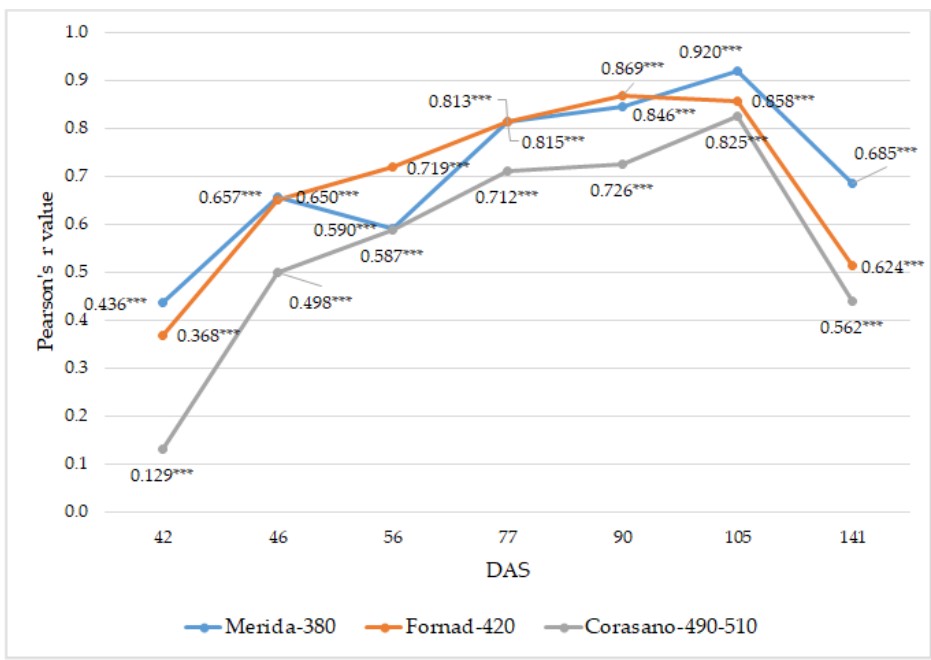

**Figure 11.** Pearson correlation coefficient (r) dynamics among NDVI$_{UAV}$ and yield t ha$^{-1}$ as a result of different hybrids (2021, Debrecen). Note: ***. Correlation is significant at the 0.001 level.

Examining the effect of nutrient levels on NDVI$_{UAV}$ values (Figure 12) at different measurement times, the difference was statistically confirmed ($p < 0.001$). For all DAS, the 160 kg ha$^{-1}$ N dose resulted in higher NDVI values. A nutrient level of 80 kg ha$^{-1}$ N resulted in 6.5% lower values in the DAS average, while the control was 15.5% lower.

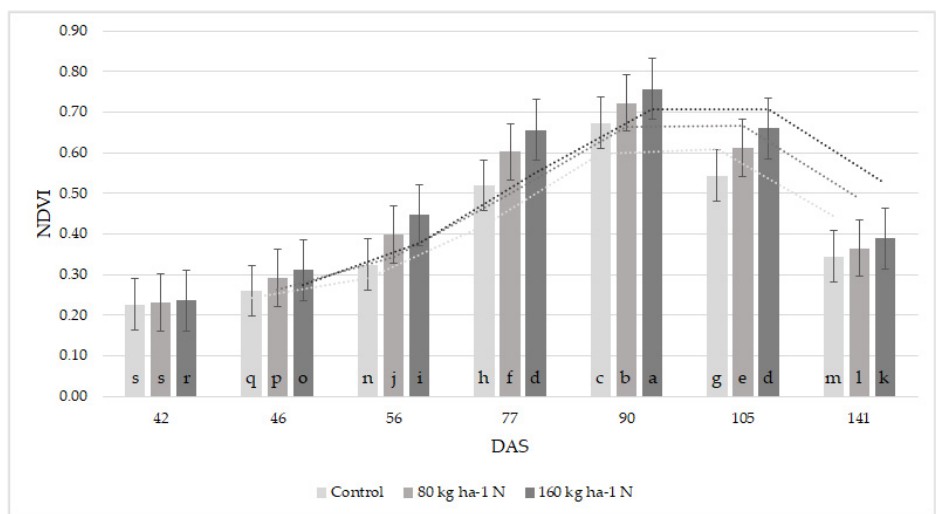

**Figure 12.** The NDVI$_{UAV}$ values of maize as the effect of fertilization treatments at different DAS (Debrecen, 2021); means of the varieties ± standard error. LSD = 0.005.

The Pearson correlation values (r) show a large variation with N ha$^{-1}$ fertilization in Table 6.

**Table 6.** Pearson correlation coefficient (r) values among NDVI$_{UAV}$ and yield t ha$^{-1}$ as a result of N ha$^{-1}$ fertilization (2021, Debrecen).

| DAS | Group | Variables | Pearson's r | 95% Cl Lower | 95% Cl Upper | N |
|-----|-------|-----------|-------------|--------------|--------------|---|
| 42 | Control | NDVI$_{UAV}$ - Yield t ha$^{-1}$ | −0.058 *** | −0.219 | 0.107 | 144 |
| | 80kg N ha$^{-1}$ | NDVI$_{UAV}$ - Yield t ha$^{-1}$ | 0.514 *** | 0.383 | 0.625 | 144 |
| | 160kg N ha$^{-1}$ | NDVI$_{UAV}$ - Yield t ha$^{-1}$ | 0.149 *** | −0.015 | 0.305 | 144 |
| 46 | Control | NDVI$_{UAV}$ - Yield t ha$^{-1}$ | 0.258 *** | 0.099 | 0.405 | 144 |
| | 80kg N ha$^{-1}$ | NDVI$_{UAV}$ - Yield t ha$^{-1}$ | 0.623 *** | 0.511 | 0.714 | 144 |
| | 160kg N ha$^{-1}$ | NDVI$_{UAV}$ - Yield t ha$^{-1}$ | 0.268 *** | 0.110 | 0.414 | 144 |
| 56 | Control | NDVI$_{UAV}$ - Yield t ha$^{-1}$ | 0.179 *** | 0.016 | 0.333 | 144 |
| | 80kg N ha$^{-1}$ | NDVI$_{UAV}$ - Yield t ha$^{-1}$ | 0.561 *** | 0.438 | 0.664 | 144 |
| | 160kg N ha$^{-1}$ | NDVI$_{UAV}$ - Yield t ha$^{-1}$ | 0.224 *** | 0.063 | 0.374 | 144 |
| 77 | Control | NDVI$_{UAV}$ - Yield t ha$^{-1}$ | 0.550 *** | 0.425 | 0.655 | 144 |
| | 80kg N ha$^{-1}$ | NDVI$_{UAV}$ - Yield t ha$^{-1}$ | 0.704 *** | 0.610 | 0.778 | 144 |
| | 160kg N ha$^{-1}$ | NDVI$_{UAV}$ - Yield t ha$^{-1}$ | 0.801 *** | 0.734 | 0.853 | 144 |
| 90 | Control | NDVI$_{UAV}$ - Yield t ha$^{-1}$ | 0.563 *** | 0.440 | 0.666 | 144 |
| | 80kg N ha$^{-1}$ | NDVI$_{UAV}$ - Yield t ha$^{-1}$ | 0.721 *** | 0.633 | 0.792 | 144 |
| | 160kg N ha$^{-1}$ | NDVI$_{UAV}$ - Yield t ha$^{-1}$ | 0.566 *** | 0.444 | 0.668 | 144 |
| 105 | Control | NDVI$_{UAV}$ - Yield t ha$^{-1}$ | 0.535 *** | 0.407 | 0.643 | 144 |
| | 80kg N ha$^{-1}$ | NDVI$_{UAV}$ - Yield t ha$^{-1}$ | 0.648 *** | 0.542 | 0.734 | 144 |
| | 160kg N ha$^{-1}$ | NDVI$_{UAV}$ - Yield t ha$^{-1}$ | 0.717 *** | 0.627 | 0.788 | 144 |
| 141 | Control | NDVI$_{UAV}$ - Yield t ha$^{-1}$ | 0.225 *** | 0.063 | 0.375 | 144 |
| | 80kg N ha$^{-1}$ | NDVI$_{UAV}$ - Yield t ha$^{-1}$ | 0.299 *** | 0.142 | 0.441 | 144 |
| | 160kg N ha$^{-1}$ | NDVI$_{UAV}$ - Yield t ha$^{-1}$ | 0.296 *** | 0.139 | 0.438 | 144 |

***. Correlation is significant at the 0.001 level; confidence level (Cl) 95%.

In autumn 2020, both the 80kg N ha$^{-1}$ and 160kg N ha$^{-1}$ treatments received the same amount of nutrients, therefore the 160kg N ha$^{-1}$ nutrient level shows a lower (weak) correlation with yield at the initial developmental stages. After top-dressing in June, on day 77 after sowing, there is already a strong correlation between NDVI$_{UAV}$ measured at 160kg N ha$^{-1}$ nutrient level and yield (r = 0.801). The control plots show weak to moderate correlations throughout the growing season (r = −0.058–0.563). The 80kg N ha$^{-1}$ nutrient level shows a strong coupling at days 77 and 90 after sowing (r = 0.704; 0.721), as shown in Figure 13.

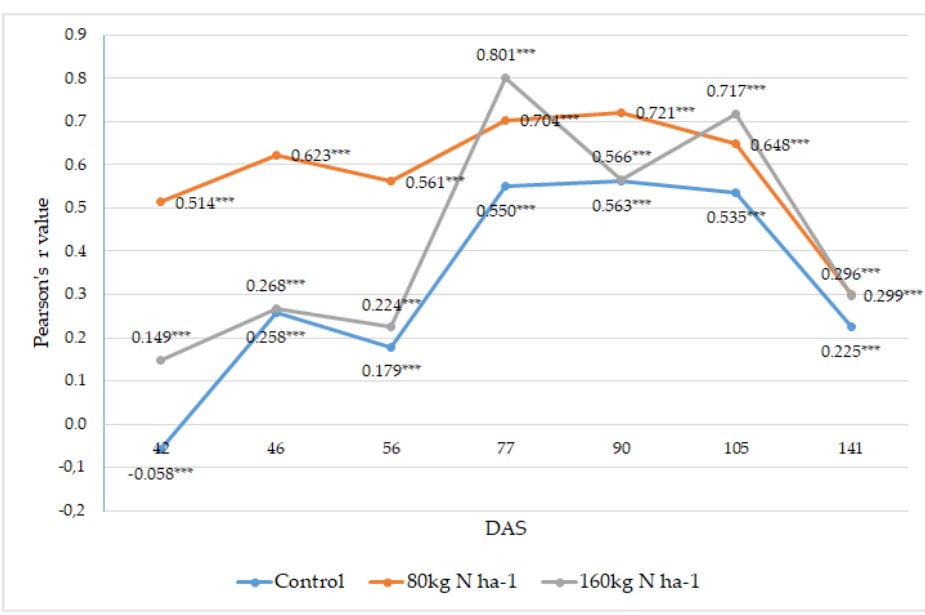

**Figure 13.** Pearson correlation coefficient (r) dynamics among NDVI$_{UAV}$ and yield t ha$^{-1}$ as a result of N fertilization (2021, Debrecen). Note: ***. Correlation is significant at the 0.001 level.

## 4. Discussion

Our UAV–based NDVI dynamics showed a similar trend to those measured by Tunka et al. [45] using satellite–based NDVI.

The NDVI dynamics of maize, averaged across cultivars and hybrids, showed a significant correlation with yield on days 90 - 105 after sowing (r = 0.821 ***–0.844 ***). The developmental stage of the plant at this time was VT and R1. According to research by Yang et al. [46], NDVI values measured at the R3 phenological state can be used to predict production, whereas our results suggest that it is more likely to be predicted at an earlier mid-vegetative stage as noted by Chivasa et al. [47], although it is highly dependent on environmental factors and agrotechnology, which may affect the accuracy of the model.

Irrigation in dry years is essential to maintain crop security. Smart nutrient management is also important for maximum yield potential. Irrigation and fertilization are treatments that ensure stability of maize yields over the years, with the potential to increase crop yields by an average of about 25% and by more than 80% in the dry season compared to the control area [48]. In the analyzed year, precipitation in June and July was well below the 30-year average, and irrigation in June significantly increased NDVI values, which can be seen on day 77 after sowing. Ihuoma et al. [49] reported similar correlations in the case of remote sensed NDVI and different measured factors by irrigation under the influence of draught. In the examined year, the yield-enhancing effect of irrigation, on average of the treatments, was 14%. In the control plots, there was a 1% increase in yield as a result of irrigation. The 80 kg ha$^{-1}$ N nutrient level resulted in a 6% increase, while the 160 kg ha$^{-1}$ N treatment resulted in a 29% yield increase. Irrigation contributed to increased yields and crop security, and Pearson's correlation was strong.

N ha$^{-1}$ doses also significantly ($p < 0.001$) increased NDVI values of maize. The overall correlation with yield showed variability, as a significant proportion of the N active ingredient was applied as liquid top-dressing in June, and therefore showed a strong correlation (r = 0.704 ***–0.801 ***) with yield from day 77 of measurement. The strength of the correlation was the highest at the V12 phenophase (DAS: 77) due to the 160 kg ha$^{-1}$ N treatment. In the case of lower dose (80 kg ha$^{-1}$ N), the correlation was more stable during the vegetation. This can be attributed to the fact that the study was conducted in the average of irrigated and non-irrigated treatments. Due to droughty weather at the end of the

vegetative phase and at the edge of the generative phase, the absorption of excess N by the plants is limited and has a negative effect.

The NDVI values measured in the tillage treatments differed statistically ($p < 0.001$) during the maize growing season. The negative effects of droughty weather in summer were confirmed in the study of NDVI dynamics of maize under conventional tillage. Tillage may be a tool for managing the effects of meteorological elements, particularly their abrupt changes in habitat, in addition to being an agricultural technology tailored to the needs of a certain crop [50]. The strongest correlation between NDVI values measured in the three tillage modes studied and yield was observed on the 105th day after sowing (r = 0.825 ***–0.920 ***).

The hybrids showed the strongest correlation at different times during the growing season, which is due to the different maturity times (FAO). The NDVI dynamics showed a similar trend for all three hybrids, but were statistically different ($p < 0.001$) among measurement times.

## 5. Conclusions

With UAV-based remote sensing, more accurate information can be gathered about plant condition, because atmospheric disturbing factors can be excluded and phenological phases can be precisely determined.

Fertilization, tillage, hybrid and maize plant phenological stage (DAS) influenced NDVI$_{UAV}$ values. In conclusion, based on our results, NDVI images taken with the help of UAVs showed variable correlation with maize crop yield under different cropping technologies.

The NDVI dynamics suggest that tolerance to environmental stress factors of hybrids of different genotypes can be increased by different combinations of N ha$^{-1}$ and constant PK treatments, and by irrigation in response to climate change.

The promotion of environmentally friendly tillage techniques is an important aspect of sustainability, and we recommend the use of ripping and strip-tillage over winter ploughing.

In a droughty year, the higher-FAO-number hybrids (Corasano FAO-490-510; Fornad FAO-420) proved to be more economical than the early–maturing hybrids (Merida FAO-380).

In the last few years in our region, irrigation has played a prominent role in terms of crop safety. Only a few areas have the possibility of irrigation; therefore, water-saving tillage methods and reasonable nutrient supply are insufficient from the point of view of sustainability.

The accuracy of the NDVI–yield correlation can be further enhanced by analyzing several years of data, which is the priority for us in the future.

**Author Contributions:** Conceptualization, A.T., E.K., and P.R.; methodology, A.T. and P.R.; software, P.R.; validation, A.T., L.R., and É.H.; formal analysis, E.K. and C.J.; investigation, A.T., P.R., and L.R.; resources, T.R. and A.T.; data curation, A.T., É.H., and T.R.; writing–original draft preparation, A.T., P.R., E.K., and É.H.; writing–review and editing, E.K., C.J., P.R., and A.T.; visualization, A.T. and P.R.; supervision, A.T. and P.R.; project administration, A.T.; funding acquisition, A.T. All authors have read and agreed to the published version of the manuscript.

**Funding:** Project no. TKP2021-NKTA-32 has been implemented with the support provided by the Ministry of Innovation and Technology of Hungary from the National Research, Development and Innovation Fund, financed under the TKP2021-NKTA funding scheme.

**Data Availability Statement:** The data presented in this study are available on request from the corresponding author. The data are not publicly available due to institutional policy.

**Conflicts of Interest:** The authors declare no conflicts of interest. The funders had no role in the design of the study; in the collection, analyses, or interpretation of data; in the writing of the manuscript; or in the decision to publish the results.

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
