# Peer review of "Assessment of NDVI Dynamics of Maize (Zea mays L.) and Its Relation to Grain Yield in a Polyfactorial Experiment Based on Remote Sensing"

_agriculture, doi:10.3390/agriculture13030689_

Round 1
Reviewer 1 Report
Assessment of NDVI dynamics of maize (Zea mays L.) and its relation to grain yield in a polyfactorial experiment based on remote sensing:
· Add/Replace the name of the study area with the Keywords.
· In the last paragraph of the Introduction, the authors should mention the weak point of former works (identification of the gaps) and describe the novelties of the current investigation to justify that the paper deserves to be published in this journal.
· “Irrigation and fertilisation are treatments that ensure stability of maize yields over the years, with the potential to increase 328 crop yields by an average of about 25% and by more than 80% in dry season”. Explain.
· Discuss more the Pearson correlation coefficient (r) dynamics among NDVIUAV and Yield t ha-1 as a result of N fertilization.
· Focus on the advantages/disadvantages of the proposed method concerning the obtained results.
· At the end of the manuscript, explain the implications and future works considering the outputs of the current study.
· The quality of the language needs to be improved for grammatical style and word use.
Author Response
Dear Reviewer!
The responses can be found in the attached pdf.
Kind regards:
Authors

Reviewer 2 Report
This manuscript gave a systematic analysis on NDVI daynamics and yields of three varieties of maize. The experiment is well disigned, and the acquired data are sufficient. But more analysis is needed to better support authors' viewpoints. Some specific suggestions are listed below:
1. the introduction part needs to be improved in logic.
i.g. (a) Line 46-49 is kind of a repeat of Line 29-30, which are both describing the importance of maize crop.
(b) The two sentances in Line 56-58 seems repeating the same idea.
(c) Line 59, ML is mentioned here for yield estimation, but this reseach only used correlation analysis to reveal the correlations between NDVI and yield under different conditions.
I think authors should reorganize the introduction part to give readers clearer objectives on what this research is going to do.
2. Line 66-69: The two hypotheses seem like expressing the same thing? and they have been already proved in previous researches (Line 63-65). and those two hypotheses were put forward so suddenly that there is no necessary explanation.
3. Line 303-317: It's more suitable to put those two paragraghs in introduction part.
4. the authors emphasized the impact of climate change on crop production, but in this reseach, only the factor of irrigation is related to climate change, and only the correlations between the NDVI and yield under different two kind of irrigation groups were analyzed. We don't know how the water imput amount influence the yield? the authors can add the stastical data on yield.
5. The discussion part can be improved. i.g., Line 327-328, "Irrigation and fertilisation are treat-327 ments that ensure stability of maize yields over the years", and Line 334: "Irrigation has contributed to increased yields and crop security." The data can not support this point because no yield data were provided.
6. The conclusions also need to be revised.
(a) The first paragragh is suggested to be put in discussion part.
(b) All the conclusions should be directly discussed in previous part based on the results. because there was no yield data to support your point, we can only know that some of the factors are good to improve the correlations between NDVI and yield, not the yield itself.
(c) New information should not be appeared in this part (i.g. the last paragragh).
7. Check the English grammar, i.g.
Line 118 "Hannover software." missing information.
Line 167 "It was expected to find a close correlation was expected (Figure 4.)" check the grammar.
and Line 259-261
Author Response

(The authors gave the same response as above.)

Reviewer 3 Report
The maize NDVI dynamics and its in relation to grain yield in a polyfactorial experiment based on remote sensing. My comments regarding the development of the work are as follows:
1- The introduction has a good literature review, citing 28 articles on the subject, including the issue of climate change and its impact on maize production. However, the experiment location figure is not very good or suitable for someone who is not familiar with Hungary. It is necessary to improve the map, regarding its macro location (Europe).
2- The characteristics of the meteorological station for data acquisition are not mentioned in section 2.3. Is it a weather station only for the experiment? Is it a conventional weather station of the Hungarian meteorological service? We know that an adequate climatological normal needs at least 30 years of data to characterize the climatology of a location. If this entire series is not available, it would be important to add satellite data for these variables to support the analysis. Temperature and rainfall are important, but where is the solar radiation data?
3- In addition, I think there is a problem with figure 2, as the temperature is showing an average above 60°C in some months. Is it a desert region? What is the Koppen climate classification for the location?.
4- An important association of NDVI data is with LAI data. I suggest checking the possibility of activating this variable in the analyses.
Author Response

(The authors gave the same response as above.)

Reviewer 4 Report
Dear Authors,
Please find my comments and recommendations in the attached file!
Kind regards!

Author Response

(The authors gave the same response as above.)

Round 2
Reviewer 1 Report
I appreciate the authors addressing the comment. The manuscript can be accepted in its current form. Congrats!
Reviewer 2 Report
The authors have addressed all the problems and revised the text as suggested. It's recommended for publication.